

# Interthalamic adhesion size in aging dogs with presumptive spontaneous brain microhemorrhages: a comparative retrospective MRI study of dogs with and without evidence of canine cognitive dysfunction

Curtis W. Dewey[1,2,3], Mark Rishniw[1], Philippa J. Johnson[1],
Emma S. Davies[1], Joseph J. Sackman[2], Marissa O'Donnell[2], Simon Platt[4]
and Kelsey Robinson[4]

[1] Clinical Sciences, Cornell University College of Veterinary Medicine, Ithaca, NY, USA
[2] Long Island Veterinary Specialists, Plainview, NY, USA
[3] Rochester Veterinary Specialists and Emergency Services, Rochester, NY, USA
[4] Department of Small Animal Medicine and Surgery, University of Georgia, Athens, GA, USA

Corresponding author
Curtis W. Dewey,
cwd27@cornell.edu

## ABSTRACT

**Objective:** Spontaneous brain microhemorrhages in elderly people are present to some degree in Alzheimer's disease patients but have been linked to brain atrophy in the absence of obvious cognitive decline. Brain microhemorrhages have recently been described in older dogs, but it is unclear whether these are associated with brain atrophy. Diminution of interthalamic adhesion size-as measured on MRI or CT-has been shown to be a reliable indicator of brain atrophy in dogs with canine cognitive dysfunction (CCD) in comparison with successfully aging dogs. We hypothesized that aging dogs with brain microhemorrhages presenting for neurologic dysfunction but without obvious features of cognitive decline would have small interthalamic adhesion measurements, like dogs with CCD, compared with control dogs.
The objective of this study was to compare interthalamic adhesion size between three groups of aging (>9 years) dogs: (1) neurologically impaired dogs with presumptive spontaneous brain microhemorrhages and no clinical evidence of cognitive dysfunction (2) dogs with CCD (3) dogs without clinical evidence of encephalopathy on neurologic examination (control dogs). MR images from 52 aging dogs were reviewed and measurements were obtained of interthalamic adhesion height (thickness) and mid-sagittal interthalamic adhesion area for all dogs, in addition to total brain volume. Interthalamic adhesion measurements, either absolute or normalized to total brain volume were compared between groups. Signalment (age, breed, sex), body weight, presence and number of SBMs, as well as other abnormal MRI findings were recorded for all dogs.
**Results:** All interthalamic adhesion measurement parameters were significantly ($P < 0.05$) different between control dogs and affected dogs. Both dogs with cognitive dysfunction (12/15; 80%) and dogs with isolated brain microhemorrhages had more microhemorrhages than control dogs (3/25; 12%). Affected dogs without cognitive dysfunction had significantly more microhemorrhages than dogs with cognitive

dysfunction. In addition to signs of cognitive impairment for the CCD group, main clinical complaints for SBM and CCD dogs were referable to central vestibular dysfunction, recent-onset seizure activity, or both. Geriatric dogs with spontaneous brain microhemorrhages without cognitive dysfunction have similar MRI abnormalities as dogs with cognitive dysfunction but may represent a distinct disease category.

# INTRODUCTION

Geriatric humans can suffer brain microhemorrhages, often referred to as cerebral microhemorrhages or cerebral microbleeds. Brain microhemorrhages can occur secondary to several disorders, including neoplasia, coagulopathies and hypertensive states (e.g., renal-associated hypertension) (*Jouvent, Puy & Chabriat, 2016*; *Ungvari et al., 2017*). These microhemorrhages can also occur as a result of cerebral amyloid angiopathy, which is a specific cerebral vasculopathy of geriatric humans caused by the accumulation of β-amyloid protein in vessel walls of brain arterioles and capillaries. Microbleeds are often detected by MRI in geriatric human patients with a variety of underlying brain disorders, including Alzheimer's disease and cerebral amyloid angiopathy and in 5–15% of geriatric humans, as incidental findings (*Jouvent, Puy & Chabriat, 2016*; *Ungvari et al., 2017*; *Murao, Rossi & Cordonnier, 2013*; *Bos et al., 2018*; *Sharma et al., 2018*). Microbleeds in humans appear on MRI as circular, ovoid or "dot-like" parenchymal lesions, best identified with T2* gradient echo or susceptibility-weighted MR sequences. Although there is some variability in the literature concerning what constitutes a microhemorrhage, most reports include lesions that are less than 5.0 mm in diameter (*Jouvent, Puy & Chabriat, 2016*; *Ungvari et al., 2017*; *Murao, Rossi & Cordonnier, 2013*; *Bos et al., 2018*; *Sharma et al., 2018*). Pathologists have identified both cerebral amyloid angiopathy and cerebral microbleeds in geriatric dogs (*Uchida, Nakayama & Goto, 1991*; *Shimada et al., 1992*; *Wegiel et al., 1995*; *Yoshino et al., 1996*; *Colle et al., 2000*; *Jakel et al., 2017*; *Rodrigues et al., 2018*), and clinicians have described putative microhemorrhages in geriatric dogs undergoing MRI (*Fulkerson et al., 2012*; *Hodshon, Hecht & Thomas, 2014*; *Kerwin et al., 2017*).

Anecdotally, we have observed geriatric dogs with and without evidence of cognitive dysfunction occasionally present with recent onset seizure activity and/or central vestibular dysfunction. The vestibular dysfunction in these dogs is generally much milder than that displayed in dogs with "geriatric vestibular syndrome" and tends to improve more rapidly than the peripheral disorder. These dogs typically exhibit varying numbers of punctuate lesions on MRI, consistent with microbleeds or microhemorrhages described in humans. As with human patients with spontaneous brain microhemorrhages, veterinary neurologists often fail to identify any metabolic cause for microhemorrhages in geriatric dogs. In one study of putative microhemorrhages in dogs, an association was

found between systemic hypertension and the presence of brain microhemorrhages (*Kerwin et al., 2017*); however, systemic hypertension can be a result of brain pathology as well as causing brain pathology, as the control center for systemic blood pressure is located in the medulla and is responsive to changes in cerebral oxygen levels (*Reis et al., 1994*).

Dogs with cognitive dysfunction have smaller interthalamic adhesion heights than similarly aged control dogs without cognitive dysfunction (*Hasegawa et al., 2005*; *Noh et al., 2017*). This change is not considered specific for CCD but is indicative of a generalized neurodegenerative process (*Hasegawa et al., 2005*).

We hypothesized that aging dogs with cognitive dysfunction and dogs presenting with acute neurological signs that cannot be attributed to intracranial neoplasia, hypertensive or coagulopathic intracranial hemorrhage, but without evidence of cognitive dysfunction, often have lesions consistent with spontaneous brain microhemorrhages detected on MRI, compared to similarly aged dogs without encephalopathy or cognitive dysfunction. We further hypothesized that dogs with cognitive dysfunction, and dogs without cognitive dysfunction but with evidence of spontaneous brain microhemorrhages, would have smaller interthalamic adhesion height and area than similarly aged control dogs.

## MATERIALS AND METHODS

We identified the dogs for our study using three strategies. For the control dogs, we searched medical records for older (>9 years) dogs who had brain MRIs performed but had no clinical signs of encephalopathy on neurologic examination (e.g., nasal disease, peripheral vestibular disease). Because few geriatric dogs without clinical evidence of brain disease undergo brain MRI, we acquired additional control geriatric brain MR images from two sources: 10 mixed-breed retired sled dogs with normal neurologic examinations that had been imaged as part of another study and six neurologically normal small-breed dogs whose owners volunteered for a no-cost brain MRI prior to scheduled dentistry procedures. In addition to being 9 years or older, control dogs were included if they had no historical or clinical evidence suggesting structural brain disease and no conditions that might predispose to spontaneous brain hemorrhages (e.g., hypertension, coagulopathy).

For dogs with cognitive dysfunction, we searched medical records for dogs with a clinical diagnosis of cognitive dysfunction that had undergone MRI. We based our diagnosis of cognitive dysfunction on previously established historical and clinical findings in association with characteristic MRI abnormalities (*Landsberg, Nichol & Araujo, 2012*; *Dewey et al., 2019*; *Dewey, 2016*; *Schutt, Toft & Berendt, 2015*). We only included cases of canine cognitive dysfunction for which this diagnosis was clearly stated in the medical record and supported by the MRI report.

For dogs with spontaneous brain hemorrhages, we searched the medical records for dogs that had undergone MRI for investigation of encephalopathic signs and had an imaging diagnosis of brain microhemorrhages assigned. We defined spontaneous brain microhemorrhages using the same criteria as had been previously proposed in humans and dogs; namely, as punctate areas of signal void in the brain parenchyma of dogs (Fig. 1) with no underlying metabolic reason (e.g., coagulopathy, endocrine disorder, renal hypertension, vascular neoplasia, etc.) or associated intracranial reason (e.g., tumor,

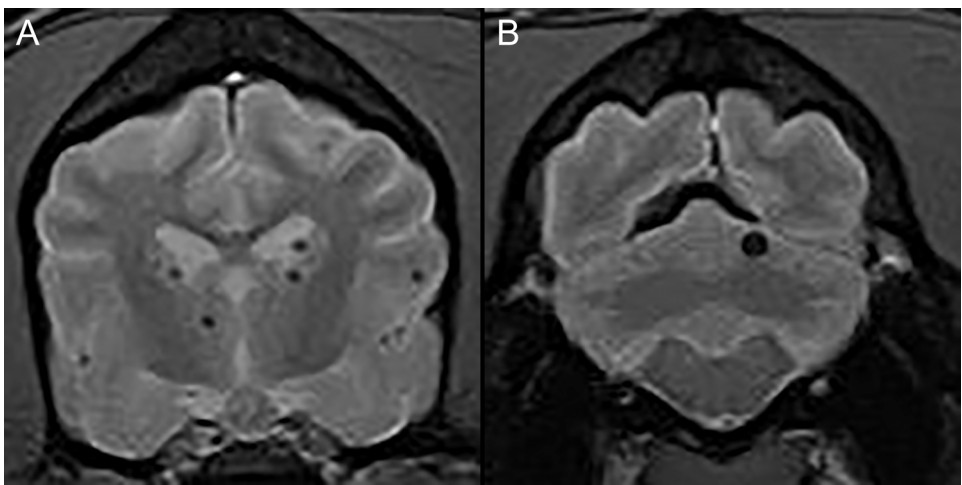

**Figure 1 Typical examples of microhemorrhages on T2\* weighted images.** Transverse T2\* gradient echo MR images demonstrating typical appearance of presumptive brain microhemorrhages. (A) Multiple microhemorrhages at the diencephalon level. (B) A single microhemorrhage in the left cerebellum.

inflammatory brain disease, etc.) for intracranial bleeding (*Sharma et al., 2018*; *Yamada, 2015*; *Boulouis, Charidimou & Greenberg, 2016*; *Kerwin et al., 2017*).

We excluded any dogs that had macrohemorrhages, or evidence of disorders that could cause brain hemorrhages, such as systemic hypertension, coagulopathies, intracranial neoplasia etc.

We searched databases from five institutions (Cornell University Hospital for Animals, Long Island Veterinary Specialists, Rochester Veterinary Specialists and Emergency Services, University of Georgia Small Animal Hospital and Oradell Animal Hospital) from 2010 to 2019.

All MRIs were performed under general anesthesia with one of six magnets: (1) 1.5 T Siemens Avanto (Munich, Germany) (2) 1.5 T Toshiba Vantage Elan (Lake Forest, CA, USA) (3) 3.0 T Philips Achieva (Nutley, NJ, USA) or (4) 3.0 T GE Discovery MR750 (Chicago, IL, USA). Imaging sequences acquired included the following: sagittal T2-weighted; transverse T2-and T1-weighted; transverse and dorsal plane T1-weighted post-gadolinium injection; transverse T2-fluid attenuated inversion recovery (FLAIR); and transverse T2\* gradient-recalled echo (GRE). For the 1.5-T MRI units, measurement parameters were as follows: slice thickness, 3.5 mm; slice gap, 3.5 mm; FOV, 185 mm; matrix size of images, 480 × 480. For the 3.0-T MRI units, measurement parameters were as follows: slice thickness, 2.0 mm; slice gap, 1.0–3.0 mm (depending on dog size); FOV, 1,101 mm; matrix size of images, 400 × 400.

For each dog, two investigators (JS and MO) measured the interthalamic adhesion height on transverse T2-weighted images, as previously described (Fig. 2) (*Hasegawa et al., 2005*). Additionally, these investigators measured the cross-sectional area of the interthalamic adhesion on mid-sagittal T2-weighted images (Fig. 3), and total brain volume for each dog, using Mimics® software. Volumetric measurements for total brain volume were performed as previously described (*Estey et al., 2017*). Both investigators used

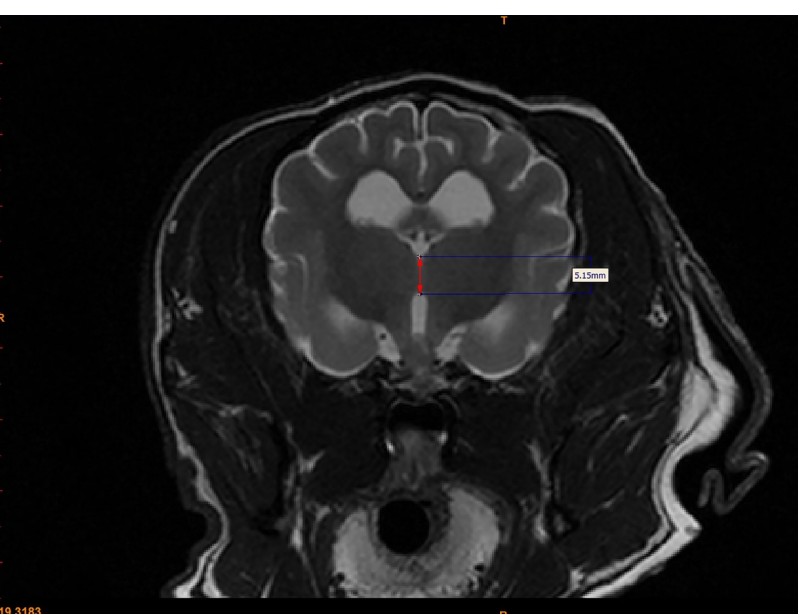

**Figure 2 Method for measuring interthalamic adhesion height from a transverse T2 weighted image.**
Transverse T2-weighted MR image depicting method of measuring interthalamic adhesion height
(thickness).                                         

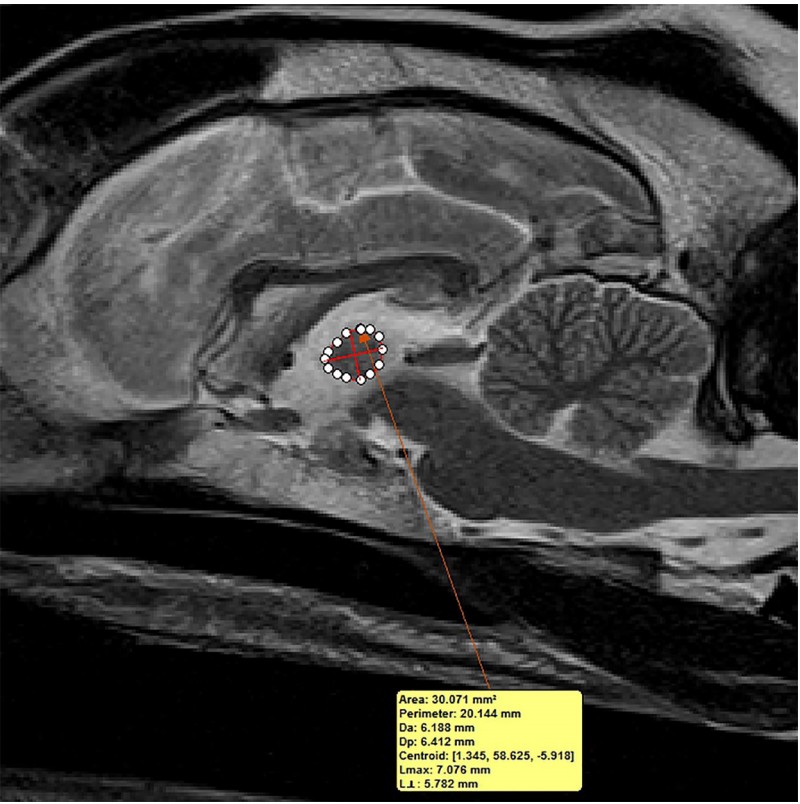

**Figure 3 Method of measuring interthalamic adhesion area from a mid-sagittal T2 weighted image.**
Mid-sagittal T2-weighted image depicting method of measuring interthalamic adhesion area.
                                                    

previously published anatomic landmarks for all measurements (*Leigh et al., 2008*) and both were blinded to the clinical status of the dogs in the study. One investigator counted all microhemorrhages evident on MR images for each dog (CD), based on transverse $T2^*$ gradient echo sequences.

### Statistical analysis

We used Kruskal Wallis tests, with subsequent multiple comparisons by the Conover method for all comparisons described below. We did not correct for any experiment-wise error rate.

We first compared ages and body weights of the three groups of dogs to demonstrate similarity of cohorts.

We compared the heights and areas of the interthalamic adhesions between the three groups of dogs both as absolute values, and after indexing to total brain volume (under the assumption that the height of the interthalamic adhesion would be proportional to the size of the brain).

Finally, we compared the number of microhemorrhages between the three groups of dogs.

To examine the reproducibility of the measurements of interthalamic adhesion height and area, one investigator re-measured these variables in 41 dogs several months after the initial measurements. We compared the pairs of measurements using a Limits of Agreement approach.

To examine the reproducibility of microhemorrhage identification, one investigator re-counted the microhemorrhages in 10 dogs with either cognitive dysfunction or spontaneous brain microhemorrhages. We examined the absolute difference between the pairs of results, but performed no statistical analyses.

## RESULTS

Our study consisted of 52 aging dogs: 25 control dogs, 12 dogs with spontaneous brain hemorrhages and neurologic impairment without evidence of cognitive dysfunction and 15 dogs with evidence of cognitive dysfunction. Control dogs were younger than affected dogs (median age 11 years vs. 13 years; $P < 0.0001$). Bodyweights of dogs in each group did not differ ($P = 0.9$). Breeds represented in the control dog group (CD) included mixed breed (12), Chihuahua (4), Dachshund (2) and one each of Yorkshire terrier, Beagle, Boston terrier, Golden Retriever, Coonhound, West Highland White terrier and Maltese. There were nine female spayed, three male castrated and two each of female and male intact dogs in the control group. Breeds represented in the spontaneous brain microhemorrhage group without cognitive impairment (SBM) included one each of Shih tzu, Labrador retriever, mixed breed Maltese, Greyhound, Shetland Sheepdog, Boston terrier, Chihuahua, Golden retriever, Pug, Yorkshire terrier and Bichon Frise. There were six female spayed dogs, five male castrated dogs and one male intact dog. Breeds represented in the cognitive dysfunction (CCD) group included Shih tzu (2), Springer spaniel (2), mixed breed (2) and one each of Labrador retriever, Chihuahua, Miniature Poodle, Shetland Sheepdog, Cockapoo, Samoyed, Miniature Schnauzer, Tibetan terrier,

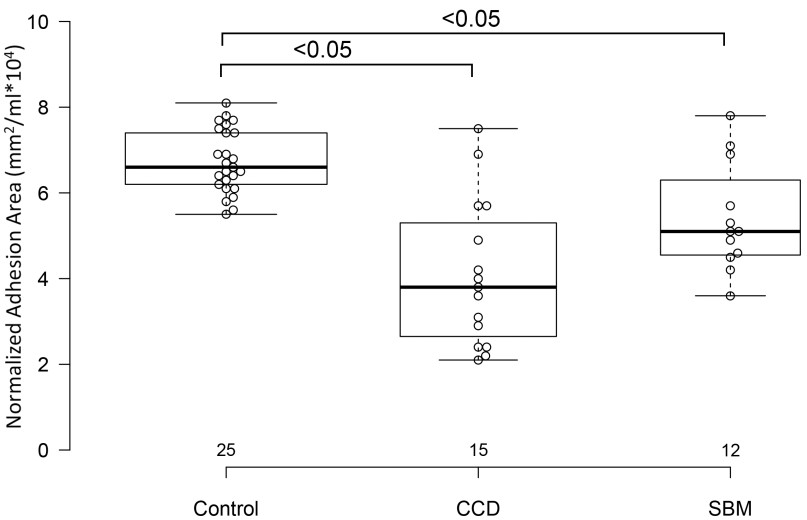

**Figure 4 Box and whisker plots of interthalamic adhesion areas normalized to total brain volume for three groups of dogs.** Box and whisker plots of interthalamic adhesion area indexed to total brain volume (TBV) for control dogs, dogs with cognitive dysfunction (CCD) and spontaneous brain microhemorrhage dogs without cognitive dysfunction (SBM).

and Wheaten terrier. There were eight female spayed, five male castrated and two male intact dogs.

Dogs with spontaneous brain microhemorrhages presented with central vestibular dysfunction (9), recent-onset seizures (2) and a combination of the two (1). Dogs with CCD presented with cognitive dysfunction alone (5), recent-onset seizures (6) central vestibular dysfunction (3) and non-ambulatory paraparesis (1).

Control dogs had taller sagittal interthalamic adhesions and larger interthalamic adhesion areas than dogs with microhemorrhages without cognitive impairment and dogs with cognitive dysfunction ($P = 0.0001$ for both comparisons). Dogs with cognitive dysfunction had taller sagittal interthalamic adhesions and larger interthalamic adhesion areas than dogs with microhemorrhages without cognitive impairment. When indexed to total brain volume, control dogs had taller sagittal interthalamic adhesions than dogs with cognitive dysfunction, but not dogs with microhemorrhages without cognitive dysfunction ($P = 0.03$). However, when indexed to total brain volume, control dogs had larger interthalamic adhesion areas than both groups of abnormal dogs ($P = 0.0001$) (Fig. 4). Dogs with spontaneous brain microhemorrhages without cognitive dysfunction and those with cognitive dysfunction did not differ from each other ($P > 0.05$).

Three control dogs (3/25; 12%) exhibited evidence of microhemorrhages, with two hemorrhages in one dog and a single hemorrhagic lesion in each of the other two. Twelve of 15 dogs with cognitive dysfunction (80%) had evidence of brain microhemorrhages. Control dogs had fewer microhemorrhages than either group of abnormal dogs. Dogs with cognitive dysfunction had fewer microhemorrhages than dogs without cognitive dysfunction ($P < 0.0001$) (Fig. 5).

When plotted against age, dogs with spontaneous brain hemorrhages and dogs with cognitive dysfunction did not display evidence of decreasing sagittal interthalamic

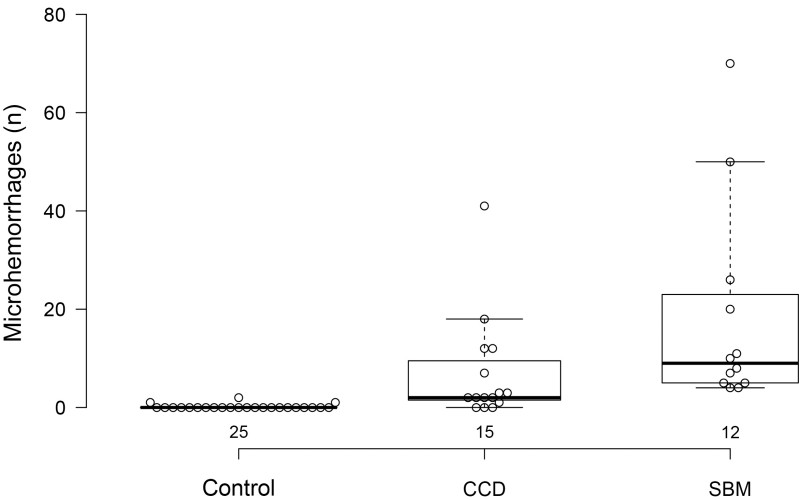

**Figure 5 Box and whisker plots of microhemorrhage numbers for three groups of dogs.** Box and whisker plots of microhemorrhage numbers for control dogs, dogs with cognitive dysfunction (CCD) and spontaneous brain microhemorrhage dogs without cognitive dysfunction (SBM).

adhesion height or interthalamic adhesion area (Fig. 6). Control dogs displayed weak evidence of decreasing interthalamic adhesion height and area with age. Dogs with cognitive dysfunction tended to have smaller interthalamic adhesion height, but not interthalamic adhesion area, than similarly aged control dogs (Fig. 6).

Duplicate measurements of interthalamic adhesion height showed no bias with 95% Limits of Agreement being ±1 mm. Similarly, duplicate measurements of interthalamic adhesion area showed no bias with 95% Limits of Agreement being ±3 mm$^2$.

Duplicate counts of microhemorrhages in 10 dogs differed by a median of one microhemorrhage, ranging from 0 to 5 (the largest difference was in a dog with 75 microhemorrhages).

## DISCUSSION

Our study demonstrates that neurologically impaired aging dogs with spontaneous brain microhemorrhages, but without evidence of cognitive dysfunction, have MRI features similar to, but less severe than dogs with cognitive dysfunction. Both groups of dogs with spontaneous brain microhemorrhages had smaller normalized interthalamic adhesion areas than similarly sized and aged control dogs. Additionally, dogs with cognitive dysfunction had smaller sagittal interthalamic adhesion heights than control dogs. These decreases appeared to be independent of age. The interthalamic adhesion measurements did not differ between the two neurologically affected groups of dogs. Somewhat surprisingly, dogs without cognitive dysfunction had more microhemorrhages than dogs with cognitive dysfunction. The apparent absence of cognitive dysfunction in the group of dogs with nearly twice the number of microhemorrhages as in the dogs with cognitive dysfunction suggests that factors other than microhemorrhages might be responsible for cognitive dysfunction.

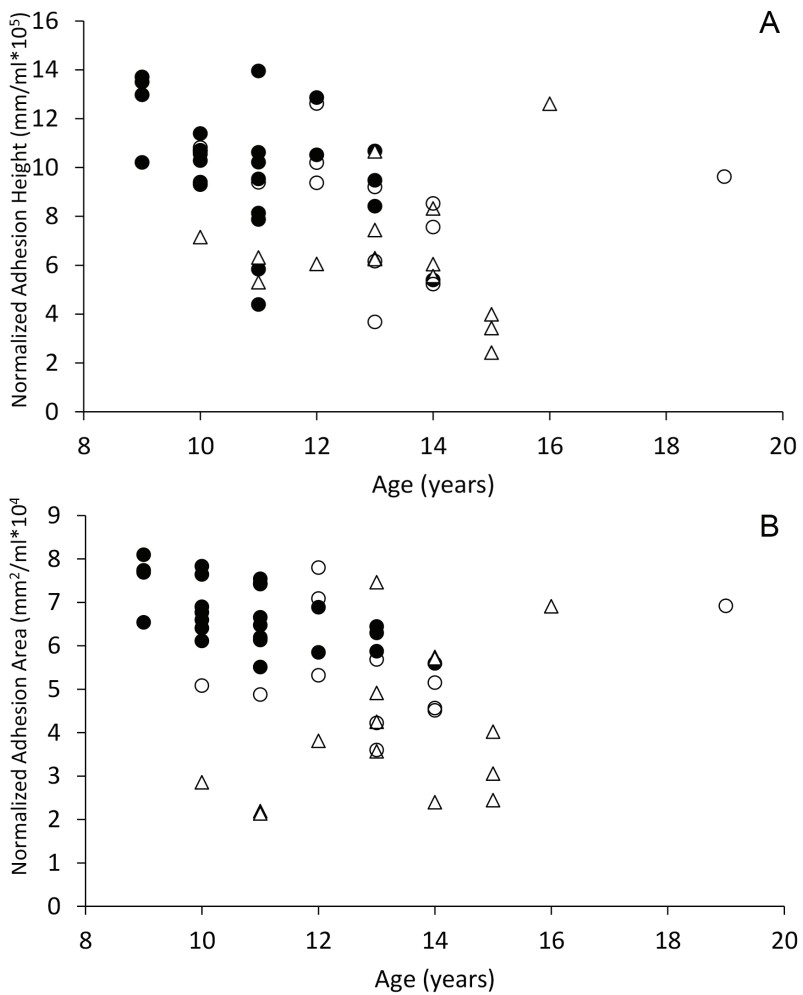

**Figure 6 Scatterplots of interthalamic adhesion measurements across age.** Scatter plots of normalized interthalamic sagittal adhesion height (A) and area (B) against age in control dogs (black circles), dogs with spontaneous brain microhemorrhages (open circles) and dogs with cognitive dysfunction (triangles).

Our results support the observations of previous investigators, who also found smaller interthalamic adhesions in dogs with cognitive dysfunction than in control dogs (*Hasegawa et al., 2005*; *Noh et al., 2017*). However, decreases in interthalamic adhesion size are not specific for cognitive dysfunction and have been reported sporadically in other disorders that can cause brain atrophy (*Hasegawa et al., 2005*). We failed to see a clear relationship with either interthalamic sagittal adhesion height or area and age in either of the affected dog groups, suggesting that these changes are not age-related.

Because we could not corroborate our findings with histopathology, we can only speculate that the lesions we observed on MRI are indeed microhemorrhages. However, the shape, size and imaging characteristics we identified are consistent with previous reports of microhemorrhages in both humans and dogs (*Jouvent, Puy & Chabriat, 2016*; *Ungvari et al., 2017*; *Murao, Rossi & Cordonnier, 2013*; *Bos et al., 2018*; *Sharma et al., 2018*; *Fulkerson et al., 2012*; *Hodshon, Hecht & Thomas, 2014*; *Kerwin et al., 2017*).

What could be causing microhemorrhages in dogs? Currently, we can only speculate about the cause. Pathologists have described histopathologic features of cerebral amyloid angiopathy in dogs and have noted the presence of cerebral hemorrhages in these cases (*Shimada et al., 1992*; *Wegiel et al., 1995*; *Yoshino et al., 1996*; *Colle et al., 2000*). Therefore, we can reasonably hypothesize that the microhemorrhages in our cohorts might be the result of cerebral amyloid angiopathy. Similar microhemorrhages in people are characteristic for cerebral amyloid angiopathy, the diagnosis of which generally relies upon MRI findings and supporting clinical features (*Sharma et al., 2018*; *Yamada, 2015*; *Boulouis, Charidimou & Greenberg, 2016*; *Smith, 2018*). Indeed, according to the Boston criteria for diagnosing human cerebral amyloid angiopathy, supportive clinical data and MRI evidence of microhemorrhages, combined with the absence of any other identifiable cause for hemorrhage, supports a diagnosis of "probable cerebral amyloid angiopathy" in the dogs in our study (*Yamada, 2015*; *Boulouis, Charidimou & Greenberg, 2016*; *Chardimou et al., 2017*; *Charidimou, 2015*; *Caetano et al., 2018*).

Putative brain microhemorrhages in dogs were described in a previous MRI-based study (*Kerwin et al., 2017*). These investigators compared dogs with microhemorrhages to all other dogs undergoing similar brain MRIs, regardless of the indications for pursuing imaging. They found that dogs with cerebral microhemorrhages were older and smaller and presented more frequently for vestibular dysfunction than the control dogs (*Kerwin et al., 2017*). Similar to this study, our populations of dogs with microhemorrhages were primarily small-breed dogs; because we selected our controls to be as similar in age as possible to the dogs with cognitive dysfunction or microhemorrhages, we cannot comment on the effect of age, although all of our dogs were older (>9 years). It is also possible that smaller breeds would be expected to predominate because they live long enough to develop degenerative brain disease, compared with larger dog breeds

Both cerebral amyloid angiopathy and cognitive dysfunction are well-established pathologies of elderly dogs (*Uchida, Nakayama & Goto, 1991*; *Shimada et al., 1992*; *Wegiel et al., 1995*; *Yoshino et al., 1996*; *Colle et al., 2000*; *Jakel et al., 2017*; *Rodrigues et al., 2018*; *Hasegawa et al., 2005*; *Noh et al., 2017*; *Landsberg, Nichol & Araujo, 2012*; *Dewey et al., 2019*; *Dewey, 2016*; *Schutt, Toft & Berendt, 2015*). Our study suggests that the dogs with spontaneous brain microhemorrhages represent a distinct neurodegenerative brain disorder with some similarities to cognitive dysfunction, but other features more reminiscent of cerebral amyloid angiopathy. Our suspicion that these dogs have cerebral amyloid angiopathy will require future histopathologic examination of brain tissue from patients with similar MRI abnormalities. Furthermore, our observations suggest that microhemorrhages are likely not the sole cause of cognitive dysfunction in dogs. Three dogs with cognitive dysfunction had no microhemorrhages and dogs without cognitive dysfunction had more microhemorrhages than dogs with cognitive dysfunction. We would have expected the opposite if microhemorrhages were causally associated.

In addition to the lack of histopathologic correlation with the imaging findings in this study, there are several other limitations to our investigation, most of which are a consequence of its retrospective nature. We made the diagnosis of cognitive dysfunction in

all dogs via historical and clinical features consistent with cognitive impairment, combined with supportive MRI findings. While this manner of diagnosing cognitive dysfunction is common practice and adequate for a clinical diagnosis, it is likely to under-diagnose dogs with mild to moderate cognitive impairment. There are a number of accurate behavioral test protocols for dogs that provide more objective data regarding cognitive health (*Landsberg, Nichol & Araujo, 2012*; *Dewey, 2016*; *Chapagain et al., 2018*; *Rofina et al., 2006*). Although the dogs with isolated microhemorrhages did not appear to have cognitive dysfunction, it is possible that they exhibited some level of cognitive dysfunction that was not appreciable without specific behavioral testing. Because microhemorrhages are known to occur in elderly people as an incidental finding, we attempted to accumulate a large cohort of aging control dogs in order to assess to what level microhemorrhages may be an incidental aging finding in dogs. In addition, when searching MRI databases for microhemorrhages, we intended to place any such incidental cases in the control group. No such cases were identified. In addition, none of the aging control dogs derived from hospital MRI records exhibited evidence of microhemorrhages. It is possible that the some of the aging control dogs that were chosen from hospital MRI records could have biased the control group and underestimated the extent of incidental microhemorrhages. However, 16 control dogs with no evidence of brain dysfunction were recruited for this investigation and only 3 (19%) exhibited evidence of microhemorrhages, all 3 of whom had 1 (2 dogs) or 2 (1 dog) lesions. Ten of the geriatric control dogs were kennel-housed and therefore not part of a home environment, unlike the remainder of the control dogs and all dogs with microhemorrhages. As such, cognitive dysfunction in some of these kennel-housed control dogs could have gone unrecognized. Two of these dogs did have small interthalamic adhesions, despite being deemed as cognitively normal. Ideally, all the control dogs would have been from home environments, in which subtle behavioral changes could have been observed by owners. Although measuring interthalamic adhesion thickness from a transaxial MR image slice is an accurate and easily applicable clinical tool for the diagnosis of CCD (*Hasegawa et al., 2005*; *Noh et al., 2017*), some inherent error in this method exists. The image slice that appears to have the largest interthalamic adhesion measurement is chosen from the slices available, which introduces a level of variability in the resultant data. Since the mid-sagittal interthalamic adhesion area is more consistent, this is likely a more accurate mode of measuring the interthalamic adhesion (*Hasegawa et al., 2005*). We found no differences between the two groups of neurologically affected dogs, but the sample sizes were small and might have failed to detect a difference.

Our main goal with this investigation was to compare interthalamic adhesion measurements between two abnormal aging groups of dogs: neurologically affected dogs with presumptive spontaneous brain microhemorrhages and no obvious cognitive impairment and dogs with evidence of CCD (with or without spontaneous brain microhemorrhages). We chose our aging control dogs based upon a lower age limit of 9 years of age. Although this lower limit is based on several reports dealing with CCD (*Landsberg, Nichol & Araujo, 2012*; *Schutt, Toft & Berendt, 2015*; *Rofina et al., 2006*),

the definition of what minimum age defines "aging", "older" or "geriatric" in dogs is somewhat arbitrary. There are several potential interpretations of this finding. One possibility is that both brain disorders investigated in this report are more likely to occur with increasing age. This phenomenon has been documented with CCD (*Landsberg, Nichol & Araujo, 2012*; *Dewey et al., 2019*; *Dewey, 2016*; *Schutt, Toft & Berendt, 2015*; *Salvin et al., 2010*; *Azkona et al., 2009*). In one study of putative microhemorrhages in dogs (*Kerwin et al., 2017*), affected dogs were significantly older than unaffected dogs, but an increasing tendency for brain microhemorrhages to occur with aging has not yet been established in this species. Another possibility is that a lower age limit of 9 years is too low for a definition of "geriatric" in dogs. If this is the case, interthalamic adhesion measurements may be less discriminative for degenerative brain pathology if the lower age limit for geriatric control dogs is increased. In one study evaluating interthalamic adhesion thickness as an indicator of cognitive dysfunction in dogs, 9 years of age was chosen as a lower limit for defining geriatric status. These investigators found a significant difference in interthalamic adhesion thickness between aging non-cognitively impaired and cognitively impaired older dogs (*Noh et al., 2017*).

Future investigation into geriatric dogs with spontaneous brain microhemorrhages will hopefully include correlating MR imaging findings with histopathology, as well as more objective assessment of cognitive status. In addition, comparing imaging features of larger cohorts of dogs with and without cognitive dysfunction might help discern whether these entities are distinct with respect to interthalamic adhesion measurements.

## CONCLUSION

We have associated small interthalamic adhesion size with spontaneous brain microhemorrhages in neurologically impaired dogs that lack evidence of cognitive dysfunction as well as in dogs with cognitive dysfunction. Dogs with microhemorrhages had smaller interthalamic adhesions than similarly aged, weight-matched control dogs. Dogs without cognitive dysfunction had more hemorrhages than dogs with cognitive dysfunction, suggesting that the cognitive dysfunction is not associated with hemorrhage number. Our data suggest but do not confirm the idea that spontaneous brain microhemorrhages in dogs might be a manifestation of amyloid angiopathy.

### Funding
The authors received no funding for this work.

### Competing Interests
Dr. Curtis Dewey is a paid consultant at both LIVS (Long Island Veterinary Specialists) and VSES (Veterinary Specialists and Emergency Services of Rochester). Mr. Joseph Sackman and Ms. Marissa O'Donnell are employes of LIVS.

## Author Contributions

- Curtis W. Dewey conceived and designed the experiments, performed the experiments, prepared figures and/or tables, authored or reviewed drafts of the paper, and approved the final draft.
- Mark Rishniw conceived and designed the experiments, performed the experiments, analyzed the data, prepared figures and/or tables, authored or reviewed drafts of the paper, and approved the final draft.
- Philippa J. Johnson conceived and designed the experiments, authored or reviewed drafts of the paper, and approved the final draft.
- Emma S. Davies conceived and designed the experiments, authored or reviewed drafts of the paper, and approved the final draft.
- Joseph J. Sackman performed the experiments, prepared figures and/or tables, authored or reviewed drafts of the paper, and approved the final draft.
- Marissa O'Donnell performed the experiments, prepared figures and/or tables, authored or reviewed drafts of the paper, and approved the final draft.
- Simon Platt conceived and designed the experiments, authored or reviewed drafts of the paper, and approved the final draft.
- Kelsey Robinson performed the experiments, authored or reviewed drafts of the paper, and approved the final draft.

## Animal Ethics

The following information was supplied relating to ethical approvals (i.e., approving body and any reference numbers):

This is a retrospective MRI investigation and therefore does not require IACUC approval.

## Data Availability

Raw volumetric for all dogs are available in the Supplemental Files.

## Supplemental Information

Supplemental information for this article can be found online at http://dx.doi.org/10.7717/peerj.9012#supplemental-information.

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
