# Peer review of "Interthalamic adhesion size in aging dogs with presumptive spontaneous brain microhemorrhages: a comparative retrospective MRI study of dogs with and without evidence of canine cognitive dysfunction"

_PeerJ, doi:10.7717/peerj.9012_

## Round 0.1 · original submission · Major Revisions

I agree with the thoughtful and thorough comments from the reviewers concerning this study. In addition to these concerns, the authors should:

1. Indicate how cases were selected - all dogs >8 years with a diagnosis of cognitive dysfunction or spontaneous brain microhemorrhage, or a select subgroup.
2. Discuss whether MRI is an appropriate method to detect hemorrhages in brain or rather reflects surges of blood supply.
3. Elaborate on evidence that thalamic adhesions, which do not involve neuronal signaling between right and left thalami, constitute a reasonable surrogate for tiny blood seepage.

Reviewer 1 ·

Basic reporting

The manuscript is well-written with clear professional English. The introduction and background sets the parallel between brain microhemorrhages in dogs and humans and states the potential association between brain microhemorrhages and dementia such as Alzheimer’s disease as well as pathologic mechanisms through cerebral amyloid angiopathy. While the association between microhemorrhages and disease states in humans in compelling and provides a rationale to undertake this study, the fact that the author do not have histopathology of the brain of the dogs included in this study is an important limitation which is acknowledged. Nevertheless, a lot of the introduction and discussion is based on the histopathological changes found in humans with microbleeds, cerebral amyloid angiopathy and/or Alzheimer disease which is not proven to happen in dogs with microbleeds based on this study as it lacks histopathology. I would recommend that the discussion is rephrased to discuss less the histopathological changes in humans and be more grounded into the results of this study.
Moreover, the authors emphasize that cerebral amyloid angiopathy may be a precursor stage to Alzheimer disease citing Ref 4. However, reference 4 is a metanalysis that showed no association between cerebral microbleeds and the risk of dementia. It should probably be emphasized more than the association between the two condition is controversial in humans.

The introduction also would benefit from being supported by more references. For example, the authors describe on line 69-73 that some geriatric dogs are presented with central vestibular dysfunction or recent onset of seizure activity, the vestibular disturbances being “much milder” and “tends to improve more rapidly” than dogs with the well described geriatric vestibular syndrome. However this is not supported by a reference and may only relate to the experience of the authors alone. Similarly, on lines 75-76, the authors claim that no metabolic cause has been identified in dogs with microbleeds although reference 17 showed an association between the presence of microbleeds and proteinuria while Lowrie et al., (vet radiol and ultrasound, 2012) observed that all dogs with putative microbleeds were hypertensive. Although this is not an identification of an absolute cause for the microbleeds these metabolic conditions should be discussed here.

Experimental design

The authors state the hypothesis at the end of the introduction. I would suggest that stating the hypothesis in the abstract would also help the reader to identify more clearly the goal of the study.
The hypothesis is: “that geriatric dogs with microhemorrhages without evidence of cognitive
dysfunction have a form of vascular -associated degenerative brain disease and that these dogs
have similar, abnormal, brain MRI features as dogs with cognitive dysfunction”. A second hypothesis is “More specifically, we hypothesized that interthalamic adhesion measurements in affected dogs would be smaller than that of aging geriatric control dogs (without evidence of encephalopathy or cognitive dysfunction).

In order to answer the first hypothesis, meaning to prove that dogs with microhemorrhages have a form of vascular associated degenerative brain disease, this would definitely require histopathology. I would recommend that the hypothesis is changed and not include any histopathology related terms. Such as “vascular-associated degenerative”. Secondly, in order to compare the MRI features of between dogs with microhemorrhages and dogs without, it would require a control group which is strictly aged-matched. However, it is stated on line 135 that control dogs are significantly younger than dogs in the other two groups. On line 180 and 311 it is stated that the control are age-matched. This would need further clarification as if the control are age-matched they cannot be significantly younger at the same time. All the significant results found in this study could actually be the results of affected dogs being older than control dogs as the interthalamic adhesion size as well as the presence of microbleeds have been associated with older age.

Also the author took a novel approach to try and distinguish between dogs with and without CCD within all dogs that present with cerebral microbleeds, the control group is more arbitrary as it includes dogs with cerebral microbleeds as well but no evidence of encephalopathy. However the absence of clinical signs is solely based on the neurological examination which may not be able to detect changes in cognition in these dogs. It would be more objective to classify the dogs as having cerebral microbleeds or not and then within these dogs divide them into having such or such clinical signs. Especially if the dogs with cerebral microbleeds tend to experience quick clinical recovery of vestibular signs as stated by the author, is it possible that the control dogs with cerebral microbleeds have had some vestibular episodes that have now resolved and are just in a later phase when evaluated clinically?
For the inclusion criteria, some studies mention a size limit of 4-5.7mm of diameter to be classified as a microbleed. Has this been used in this study?
Concerning the exclusion criteria, it is unclear in the methods if concurrent diseases that could cause vestibular signs or seizures have been ruled out in the affected dogs and if they have been ruled out, please explain using which tests.

The authors made some efforts to compare objectively their different groups. However some of the methods could be improved to make the study more reproducible and reduce bias:
On line 113, it is mentioned that two investigators have performed the measurement but it is unclear how this has been used in the study. Were the cases divided among the two? Has any inter-observer agreement been studied which would test the reproducibility of these results?
On line 120 it is mentioned that the number of microbleed was counted based on T2*GRE or SWI. However, there is a proven difference in sensibility between T2* GRE and SWI sequences may result in higher number of microhemorrhages in dogs evaluated by SWI (Guo et al., Clin Neuroradiol 2013). Can you comment on correlation between the two sequences? In order to make a rigorous investigation only one of those two sequences should be used in each dog.

Overall, the division in these 3 groups is not completely rigorous and based on subjective assessment (neurological examination which is insensitive to detect mild deficits in cognition or resolved clinical signs in diseases that improve spontaneously quickly). The way it is investigated without histopathology does not answer the primary hypothesis. A more rigorous analysis would probably involve looking at dogs with microbleeds only and compare dogs with and without CCD to investigate the relationship between these two conditions. Moreover, the control group being younger makes all of the results non-interpretable as they could be due to the age difference only.
Moreover, when looking at the raw data it seems that some dogs in the CCD group also presented for signs of central vestibular deficits and recent onset of seizures which makes the distinction between the three groups very unreliable and not clearly defined.

Validity of the findings

The lack of an appropriately age-matched control group is a major limitation preventing to reach a conclusion on the results observed.
On lines 161-164, the proportions of dogs with microhemorrhages in each group are presented which allows the construction of a contingency table for dogs with/without microhemorrhages and dogs with/without CCD which seems to show an association between presence of microhemorrages and presence of CCD. This finding would be worth mentioning and commenting.

In the Discussion section, the conclusions once again are hampered by the fact that the control group is younger and therefore all differences may be due to a difference in age between control and affected groups. It is also stated on lines 184-186 that: “dogs with microhemorrhage, with or without cognitive dysfunction, might share pathophysiologic mechanisms and might be manifestations along a spectrum of severity of a common disorder”. However, as this study lacks histopathology this conclusion cannot be reached based on the findings alone as it could be two separate diseases that happen concurrently in aged dogs.

The authors try to tackle the interesting question “what could be causing microhemorrages in dogs”. However this is not the stated purpose of this study, therefore the speculations provided in this section are not grounded in the results of this study and no results of this study suggest any potential cause. The reasoning used in the discussion, although interesting is based only on concurrent observation of the presence of hemorrhage on histopathology (and not MRI) and the presence of cerebral amyloid angiopathy in dogs which had no MRI evidence of microhemorrages similarly to this study which is therefore a weak link. In order to strengthen their argument, the authors draw an interesting parallel with the human conditions (line 206). It would be interesting to draw the parallel further by comparing which clinical data are different and which ones are similar between human and dogs with microbleeds.

The next question provided is “Could microhemorrhages in dogs be caused by cerebral amyloid angiopathy and could this, in turn, lead to cognitive dysfunction”. Similarly, with the lack of histopathology the answers to that question cannot be grounded in results from this study especially as no association between microhemorrhages and CCD is studied here (see contingency table above).
The discussion relies especially on histology features in humans which cannot be compared with the dogs of this study and this part of the discussion should therefore be removed.

One of the conclusion is that: “Our study suggests that the dogs with spontaneous brain microhemorrhages represent a distinct neurodegenerative brain disorder with some similarities to cognitive dysfunction, but other features more reminiscent of cerebral amyloid angiopathy.” This conclusion cannot be reached based on this study once again due to the lack of histopathology which would be necessary to state that this is a distinct neurodegenerative brain disorder with features of cerebral amyloid angiopathy. A disease may have many clinical signs and not all of them are always present in every case.
Similarly, the last sentence of the conclusion: “Our data support, but do not confirm the idea that spontaneous brain microhemorrhages in dogs might be a manifestation of amyloid angiopathy” cannot be supported by this study and should be removed in the absence of histopathology

Additional comments

Although this study describes a condition that has not been well described with only small case series reported, the construction of the three groups may artificially create the differences found as:
- the control group is significantly younger and the diseases studied are strongly associated with older age
- some dogs in the control group have microhemorrhages and could represent affected dogs that are not picked up on the neurological examination or affected dogs that have recovered clinically
- some dogs in the CCD group present with similar clinical signs that the dogs with microhemorrhages and no CCD which blurs the distinction between these two groups.
- Therefore, all the differences seen (or the absence of difference in between the two affected group) could be solely due to the group construction and may not have any biological relevance.

The lack of histopathology weakens this study as an important rationale used by the author is the link between microhemorrhage and cerebral amyloid angiopathy as is seen in humans.

On a side note, the consent form provided is not anonymous and contains private information from one of the participant.

In conclusion, this study does not provide a significant contribution to what is currently known about cerebral microhemorrhage as designed due to the important flaws in study design presented in this review. It may benefit from being reported as a case series of dogs with microhemorrhages. Some interesting information is present within this case series such as that the number of microhemorrhage (provided that it is counted only in T2*GRE or SWI for every dog and not one or the other) does not correlate with signs of CCD. Another interesting piece of information would be if location of the microbleed correlates with the presence of vestibular signs or recent onset of seizures as it is currently unknown if the microbleeds are actually the cause of these signs.

Reviewer 2 ·

Basic reporting

The paper is well written. Investigation of the similarities between canine and human ageing processes are of great importance and interest.
Nevertheless, the scientific or clinical merit of the data as they are presented in this manuscript are not clear.
Microhemorrhages / microbleeds were previously described in the veterinary literature. Risk factors, possible causes, related pathological conditions and outcome were also discussed in a well controled study (ref 17) pointing out the fact that these microhemorrhages are increased in number with age.
The use of interthalamic adhesion size as a marker for brain atrophy is well established as w ell (ref 19 and others) and was also demonstrated to be negatively correlated to age and significantly reduced in dogs with cognitive dysfunction.

Experimental design

In this manuscript, the authors collected data from dogs diagnosed with microhemorrhages with and without clinical signs of cognitive dysfunction and compared them to control dogs. The inclusion criteria used to allocate each dog to its study group is not always clear.

Moreover, comparing the three groups using same parameters that were used as inclusion criteria (eg -presence of microhemorrhages, brain atrophy) is not recommended.
The diagnosis of CCD is usually based on clinical signs and supported by typical MRI findings such as brain atrophy. I assume that the control group of dogs were dogs with normal cognitive function and normal MRI, therefore the idea of comparing the size of interthalamic adhesion or the presence or lack of microhemorrhages between these two group of dogs is not proper.


Material and methods:
1. Inclusion criteria are not clear nor is the “search database from three institutions". – please clarify, how were dogs recruited to each of the study group. Please include the inclusion and exclusion criteria. (It seems that the inclusion / exclusion criteria of some of the groups were then used to compare the three groups (eg. Intra-thalamic adhesion area and size and the presence of micro-bleeds)

2. Line 95 -How were the cognitive dysfunction diagnosed? please specify.

3. Please provide reference of the definition of geriatric as dogs above 8 years?

4. The differences between group 2 and 3 is not clear. Group 2 are dogs with microhemorrhages and no clinical signs. And group 3 should be no clinical abnormalities and no microhemorrhages on MRI? Please clarify were the MRI looked at before allocating dogs to the group? Was that a double blind study?

5. Control: no age, weight- and breed-matched control were available.
6. No consideration of breed and dog size is used for the statistical analysis although these were found to be significant risk factor in one of the references quoted (Kerwin SC et al 2017, ref 17)

Validity of the findings

1. The control group is not suitable if they are significantly younger than the affected dogs since we are looking at age related processes.

2. Breed of dogs are given in each group but there were not matched even though small breed were found to have significantly higher chances to suffer microhemorrhages than larger breed of dogs.

3. Figures 5, 6,7 & 8 present the same findings. Two figures should be enough. One that correlate height of interthalamic adhesion to its area and the other figure that uses either one of these parameters with or without normalisation (using the total brain volume) to correlate with number of microhemorrhages found.

4. Figure 9: Again, if control dogs were included based on normal MRI findings it is only expected that they will have significantly less microhemorrhages than both groups of affected dogs.

Additional comments

• Not clear what the objective is. What is the importance of comparing these three groups of dogs? They state that the objective is to compare these three group of dogs but the logic of doing this is not obvious and not provided.
• The definition of geriatric dogs was previously discussed (Fortney WD. Implementing a successful senior/geriatrichealth care program for veterinarians, veterinary technicians, andoffice managers. Vet Clin North Am Small Anim Pract 2012; 42:823–834 ) were size of dog is taken under consideration when determining “geriatric” in dogs. Ageing dogs maybe a better term for 8 year and older rather than geriatric.

---

## Round 0.2 · accepted · Accept

You have carefully addressed the reviewers' concerns, and aligned the conclusions more stringently with the methods and results, which has strengthened and focused the manuscript.